# The Survival Outcomes of Patients Requiring Prolonged Mechanical Ventilation

**DOI:** 10.3390/medicina59030614

**Published:** 2023-03-20

**Authors:** Chienhsiu Huang

**Affiliations:** Department of Internal Medicine, Dalin Tzu Chi Hospital, Buddhist Tzu Chi Medical Foundation, NO. 2, Min-Sheng Road, Dalin Town, Chiayi County 62247, Taiwan; hgssport@yahoo.com.tw; Tel.: +886-9-21552418

**Keywords:** prolonged mechanical ventilation, respiratory care center, five-year survival rate, successfully weaned from invasive mechanical ventilation, COVID-19-associated respiratory failure

## Abstract

*Background and objectives:* Patients who require prolonged mechanical ventilation typically experience poor long-term survival outcomes. However, there have been few studies conducted to investigate the five-year survival rate of these patients. This study aims to determine the factors that affect the five-year survival rate of patients who require prolonged mechanical ventilation, with the goal of improving their survival outcomes. *Materials and Methods:* The current retrospective, single-center study included all patients who required prolonged mechanical ventilation over a period of six years. We collected data on their age, sex, causes of acute respiratory failure leading to prolonged mechanical ventilation, comorbidities, receipt of a tracheostomy or not, weaning status, discharge conditions, and long-term outcomes. *Results:* The study examined the long-term outcomes of 403 patients who required prolonged mechanical ventilation until December 2018. Of the study population, 157 patients were successfully weaned from prolonged mechanical ventilation and discharged, 186 patients passed away in the hospital, and 60 patients remained ventilator-dependent. For all 403 patients receiving prolonged mechanical ventilation, the one-year and five-year survival rates were 24.3% and 14.6%, respectively. Among the 243 patients who were successfully weaned from prolonged mechanical ventilation, the corresponding rates were 32.6% and 21.0%. For the 157 discharged prolonged mechanical ventilation patients, the one-year and five-year survival rates were 50.3% and 32.6%, respectively. For the 60 ventilator-dependent patients, the one-year and five-year survival rates were 31.7% and 13.2%, respectively. The study revealed that successfully weaned from invasive mechanical ventilation and the receipt of a tracheostomy were influential factors in the five-year survival rate of patients requiring prolonged mechanical ventilation. *Conclusions:* Patients requiring prolonged mechanical ventilation may experience poor survival outcomes. Nevertheless, two key factors that can improve their long-term survival are successfully weaned from invasive mechanical ventilation and receipt of a tracheostomy.

## 1. Introduction

Patients receiving prolonged mechanical ventilation (PMV) are those who require mechanical ventilation for a minimum of six hours per day and for a minimum of 21 consecutive days [1]. In my report on the clinical experience of 574 patients requiring PMV, I found a poor one-year survival rate (24.3%) [2]. According to Carson, patients who require PMV had poor long-term outcomes that did not significantly improve in 2006 [3]. In 2012, Carson reported that the one-year survival rate of PMV patients was 52%, which is an insufficient increase in survival [4]. According to the research by Damuth et al. the overall one-year survival rate of weaning ventilator units was 45.2% [5].

In Taiwan, ventilator-dependent patients encompass mechanical ventilator care in four settings: the intensive care unit, the respiratory care center (RCC) (the weaning center in the acute care hospital), the respiratory care ward (RCW) (the long-term care ward for ventilator-dependent patients), and home care services (patient is cared by caregivers or nursing home worker) [6]. However, few studies have explored the five-year survival rate of patients requiring PMV. Therefore, this study aims to determine the factors that affect the five-year survival rate of patients who require prolonged mechanical ventilation, with the goal of improving their survival outcomes.

In 2019, I provided a comprehensive report on the clinical experience of patients requiring PMV [2]. We defined a patient who was considered successfully weaned as someone who had not required mechanical ventilation for five consecutive days and was then transferred to the general ward. The successfully weaned patients requiring PMV were classified into two groups: (1) ward mortality PMV patients who died in the ward before hospital discharge, and (2) discharged PMV patients who were discharged from the hospital after successfully weaned from PMV. We considered a patient to be ventilator-dependent if they were unable to be weaned off the ventilator and had to be transferred to RCW. I set out to evaluate the long-term survival of these patients based on data collected until the end of 2018. The data were collected prior to COVID. I suggest that long-term follow-up of ventilated COVID patients is an interesting topic that is beyond the scope of this article.

Since the coronavirus disease (COVID-19) was discovered in December 2019, it has affected millions of people worldwide. COVID-19 symptoms can range from asymptomatic to respiratory failure and even death. The risk of mortality is particularly high in severe cases of hypoxic respiratory failure caused by COVID-19 pneumonia. A study by Auld et al. revealed that among 217 critically ill COVID-19 patients, the mortality rate was 35.7% (59/165) for those who required mechanical ventilation, with 4.8% (8/165) still requiring ventilation at the time of the study. Mortality was significantly impacted by older age, chronic kidney disease, and the need for mechanical ventilation [7]. Another study by Domecq et al. reported a median duration of mechanical ventilation of 8.8 days [8]. This article briefly covers the issue of COVID-19 patients with hypoxic respiratory failure.

## 2. Methods

### 2.1. Participants

The current retrospective, single-center study included all patients who required prolonged mechanical ventilation and were admitted to the RCC between 1 January 2012, and 31 December 2017 Patients were eligible for RCC admission if they met the Taiwan national health insurance requirements [2].

### 2.2. Data Collection

Data on patients requiring PMV were collected, including age, sex, comorbidities, causes of acute respiratory failure leading to PMV, receipt of a tracheostomy or not, weaning status, discharge conditions, and long-term outcomes. Survival rates were recorded at one-year, two-year, three-year, four-year, and five-year intervals, with the five-year survival rate calculated using the Kaplan–Meier estimate of the survivor method. To assess long-term survival, I reviewed the medical records of all outpatient visits for each patient until 31 December 2018, to determine whether they had passed away, and issued a death certificate for deceased patients until 31 December 2018.

### 2.3. Outcomes Measure

The main outcome is to investigate the five-year survival rate of PMV patients using the Kaplan–Meier estimate of the survivor method. The secondary outcome is to investigate the factors that influence the five-year survival rate among all PMV patients, successfully weaned PMV patients, patients discharged after successfully weaned from patients (discharged PMV patients), and ventilator-dependent PMV patients (RCW patients).

### 2.4. Statistical Analysis

The Kaplan–Meier method was utilized to estimate the cumulative probability of survival over the long-term follow-up period for all PMV patients, those who successfully weaned from PMV, patients discharged after successfully weaned from PMV (discharged PMV patients), and ventilator-dependent PMV patients (RCW patients). Additionally, the Cox proportional hazards model was employed to determine any relationships among the survival rates of the four groups. To compare the survival rates among the four groups, the Log-rank test was utilized.

## 3. Results

Long-term follow-up data were collected for 403 PMV patients (245 men and 158 women) until 31 December 2018, with a mean age of 73.1 years. The data included 186 PMV patients who died in the hospital (86 ward patients and 100 RCC patients), 157 discharged PMV patients, and 60 ventilator-dependent patients (Table 1 and Table 2, Figure 1). The one-year, two-year, three-year, four-year, and five-year survival rates for the 403 PMV patients were 24.3%, 20.3%, 17.4%, 16.4%, and 14.6%, respectively. For the 243 patients who successfully weaned from PMV, the corresponding rates were 32.5%, 28.0%, 29.4%, 24.0%, and 21.0%, respectively. The corresponding rates for the 157 discharged PMV patients were 50.3%, 43.4%, 38.6%, 37.1%, and 32.6%, respectively. For the 60 ventilator-dependent PMV patients, the corresponding rates were 31.7%, 17.7%, 15.4%, 13.2%, and 13.2%, respectively. Kaplan–Meier analysis of five-year survival rates among the four groups is illustrated in Figure 2.

## 4. Discussion

### 4.1. The One-Year Survival Rate of PMV Patients

The one-year survival rate of 403 PMV patients in this study was 24.3%. I found that PMV patients without comorbidities (*p* = 0.002, odds ratio (OR) = 3.645, 95% confidence interval 1.607–8.266) had a better one-year survival rate. However, patients over 75 years old (*p* = 0.005, OR = 0.464, 95% confidence interval 0.270–0.795), those with end-stage renal disease (ESRD) (*p* = 0.040, OR = 0.275, 95% confidence interval 0.080–0.941), and those with four or more comorbidities (*p* = 0.021, OR = 0.180, 95% confidence interval 0.042–0.773) had poorer one-year survival rates [2]. Other studies have reported the one-year survival rate of PMV patients to range from 37.2% to 61% (Table 3) [9,10,11,12,13,14,15,16,17]. Factors associated with a poor one-year survival rate include older age, failure to wean, high Acute Physiology and Chronic Health Evaluation II (APACHE II) score, and ESRD comorbidity. Carson et al. reported that patients over 65 years old had a poorer one-year survival rate than younger patients [14], which was consistent with my finding that patients over 75 years old had a poorer survival rate. In addition, thrombocytopenia, requiring vasopressors, and ESRD were also found to be associated with poor one-year survival rates [14]. Lin et al. reported that younger age and absence of liver cirrhosis were associated with a better one-year survival rate [16], while Huang et al. found that congestive heart failure, ESRD, malignancy, and liver cirrhosis were factors related to a poor one-year survival rate. However, PMV patients who had undergone tracheostomy had a favorable one-year survival rate [17]. My findings were consistent with the variables reported in the literature regarding factors influencing the one-year survival rate of PMV patients.

### 4.2. The One-Year Survival Rate of Patients Discharged after Successfully Weaned from PMV (Discharged PMV Patients)

Several studies have shown that the one-year survival rate of patients discharged after successfully weaned from PMV varies between 44.6% and 66.9% (see Table 4) [18,19,20,21,22,23]. Factors associated with a poor one-year survival rate in these patients include older age, failure to wean, high APACHE II score, high Simplified Acute Physiology score, and low body mass index. In this study, the one-year survival rate of 157 patients discharged after successfully weaned from PMV was 50.3%. I found that patients with no comorbidities (*p* = 0.035, OR = 5.203, 95% confidence interval 1.127–24.028) and those who underwent tracheostomy (*p* = 0.005, OR = 4.439, 95% confidence interval 1.551–12.701) had better one-year survival rates. However, patients over the age of 85 had a poorer one-year survival rate (*p* = 0.004, OR = 0.028, 95% confidence interval 0.084–0.616), as did those with four or more comorbidities (*p* = 0.013, OR = 0.099, 95% confidence interval 0.016–0.608) [23].

### 4.3. Five-Year Survival Rate of PMV Patients

Limited research has investigated the five-year survival rate of patients receiving PMV. A statistical analysis of the five-year survival rate indicated a significant difference between 403 PMV patients and 243 patients who were successfully weaned from PMV (*p* = 0.001, HR = 1.374). Patients who were successfully weaned from PMV had a better five-year survival rate than those who received PMV. Therefore, successfully weaned from mechanical ventilation can impact the long-term survival of patients requiring PMV. Of the 303 PMV patients discharged from our RCC, 243 were successfully weaned and 60 remained ventilator-dependent. The group of 243 successfully weaned patients included 157 discharged PMV patients and 86 PMV patients who passed away while in the ward. The statistical analysis of the five-year survival rate showed no significant difference between the 243 successfully weaned patients and the 60 ventilator-dependent patients (*p* = 0.575, HR = 0.920). The five-year survival rate of 60 ventilator-dependent patients was similar to that of 243 successfully weaned patients. However, a significant difference was found in the five-year survival rate between 157 discharged PMV patients and 60 ventilator-dependent PMV patients (*p* < 0.001, HR = 1.965). Ventilator-dependent patients had a worse five-year survival rate than discharged PMV patients. Moreover, the long-term use of invasive ventilators did not increase the five-year survival rate of ventilator-dependent PMV patients.

### 4.4. Five-Year Survival Rate of Discharged PMV Patients

Schönhofer et al. found that the one-year and three-year survival rates for 293 discharged PMV patients were 49.4% and 38.1%, respectively [18]. Davis et al. reported higher survival rates among 457 discharged PMV patients with one-year, three-year, and five-year survival rates of 65%, 41%, and 29%, respectively [20]. Warnke et al. also found higher survival rates among 597 discharged PMV patients from a weaning center, with one-year, three-year, and five-year survival rates of 66.5%, 47.3%, and 37.1%, respectively [22]. In the present study of patients discharged after successfully weaned from PMV, I found survival rates of 50.3% at one year, 38.6% at three years, and 32.6% at five years. Comparing my results to those of other studies, I found that patients of this study had a worse five-year survival rate than Warnke’s case series, but similar rates to those reported by Schönhofer et al.

My earlier study also identified factors associated with poor survival rates among successfully weaned from PMV patients. Patients with ESRD, four or more comorbidities, and those who did not undergo tracheostomy had lower survival rates [23]. Tracheostomy was associated with a lower risk of in-hospital mortality in the study by Combes et al. [24]. Tracheostomy has several benefits for PMV patients, including a more tolerable airway, improved suctioning, the potential for oral feeding, enhanced communication, increased ambulation, and easier pulmonary toilet and oral hygiene. Therefore, tracheostomy is advised for RCC patients who cannot be weaned from the ventilator in the near future. However, the majority of patients and their families are often hesitant to undergo the procedure due to concerns about complications and scarring. In 2017, we implemented a program to address these concerns, resulting in only 37 PMV patients (9.7%) undergoing tracheostomy during the three-year study period [25]. Nonetheless, when successfully weaned PMV patients have fewer comorbidities and no ESRD, tracheostomy is strongly advised to improve their long-term survival.

### 4.5. Improving the Long-Term Survival Outcomes of PMV Patients

PMV is associated with poor long-term survival outcomes, and urgent action is needed to increase the proportion of patients who are successfully weaned from invasive ventilators. To achieve this, it is recommended that each RCC in Taiwan develop standard guidelines for ventilator weaning. Additionally, PMV patients with fewer comorbidities and no ESRD comorbidity may benefit from tracheostomy, as these patients tend to have better five-year survival rates. Several studies have shown that the use of noninvasive ventilation can improve survival rates for PMV patients. For example, Davies et al. found that their ventilator-dependent PMV patients who received nocturnal noninvasive ventilation had better one-year survival rates [20]. Similarly, Warnke et al. reported that patients who were discharged with a noninvasive mechanical ventilator had higher five-year survival rates than those discharged with an invasive mechanical ventilator [22]. Therefore, eligible PMV patients who require invasive mechanical ventilation should be switched to noninvasive mechanical ventilation to improve their long-term survival outcomes.

### 4.6. Clinical Outcomes in COVID-19 Patients Requiring Mechanical Ventilation

In a study of hospitalized COVID-19 patients in the UK, 9% required invasive mechanical ventilation [26]. A study from China reported that 2.3% of COVID-19 patients underwent invasive mechanical ventilation [8]. Gao et al. reported in their study that approximately one-fifth of hospitalized COVID-19 patients developed respiratory failure requiring mechanical ventilation [27]. Another study reported that 25% of patients with severe COVID-19 disease required mechanical ventilation [28]. According to Grasselli’s study, hospitalized COVID-19 patients requiring invasive mechanical ventilation had a significant mortality rate [29]. Between 1 January 2022, and 31 December 2022, ninety-three hospitalized COVID-19 adult patients with acute hypoxemic respiratory failure required admission to the intensive care unit in our hospital, as well as invasive mechanical ventilation. Patients with acute hypoxemic respiratory failure were divided into four groups based on their outcomes: nonsurviving in the ICU (n = 34), surviving in the ICU (n = 26), nonsurviving in RCC (n = 14), and surviving in RCC (n = 19). The incidence of PMV in COVID-19–related acute hypoxemic respiratory failure was 35.5%. The ICU mortality rate was 36.6%, the in-hospital mortality rate was 51.6%, and the RCC mortality rate was 42.4%. In a Chinese study of 191 hospitalized COVID-19 patients, 54 of the patients passed away.

Comorbidity was present in 91 (48%) patients, with hypertension (58 patients), diabetes (36 patients), and coronary heart disease (15 patients). Older age was associated with in-hospital mortality [30]. In our series, the mean age of the 48 nonsurviving patients was 76.4 years, and the mean age of the 45 surviving patients was 73.3 years. Comorbidities were present in 90 (96.8%) patients, with hypertension (48 (51.6%) patients), diabetes (45 (48.4%) patients), neurologic disease (34 (36.6%) patients), chronic kidney disease (27 (29%) patients), and coronary heart disease (19 (20.4%) patients).

Patients with COVID-19 are more likely to have bacterial secondary infections, some of which can be fatal. Rawson et al. found that 8% of 806 hospitalized COVID-19 patients experienced coinfection [31]. According to a study by Musuuza et al., 19% of COVID-19 patients developed secondary infections, which were associated with a poor outcome. The three most common bacteria identified among patients with superinfections were *Acinetobacter* spp., *Pseudomonas aeruginosa*, and *Escherichia coli* [32]. There were 33 COVID-19-related PMV patients in our series, and the most common cause for patients requiring PMV was superinfection, including 29 cases of pneumonia and 2 cases of sepsis. Among those with superinfections, the four most frequently identified bacteria were *Pseudomonas aeruginosa* (12 cases), *Acinetobacter baumannii* (10 cases), *Klebsiella pneumonia* (7 cases), and methicillin-resistant *Staphylococcus aureus* (5 cases). In a study by Melamed et al., 355 patients with COVID-19-associated respiratory failure, 86 (24%) required PMV (defined as >17 days of ventilator support), and the mortality rate was 46.5%. The mortality rates between the PMV group and in the patients requiring shorter mechanical ventilation (mortality rate: 43.1%) were not significantly different [33]. In the study of Bergman et al. 606 patients required mechanical ventilation for COVID-19 pneumonia, and the in-hospital mortality rate was 40.3%. Patients with intubations lasting longer than 21 days had a lower in-hospital mortality rate of 25.7%. In the first 21 days following intubation, the majority of mechanically ventilated patients with COVID-19 pneumonia died in the hospital [34]. There is little reported clinical experience with PMV in patients with COVID-19-associated respiratory failure in the literature. Additional studies will be conducted to provide clinical evidence and establish associations with COVID-19-associated respiratory failure patients on PMV.

#### Limitations of This Study

There are several limitations to the present study. First, the retrospective and single-unit nature of the study may limit the generalizability of my findings regarding long-term survival rates of PMV patients, discharged PMV patients, and ventilator-dependent patients. Additionally, there may be potential confounding factors that could influence the long-term survival rate of PMV patients, such as underlying comorbidities and treatment protocols. Finally, further research is needed to better understand the long-term survival of PMV patients.

## 5. Conclusions

Patients on PMV have been found to have a low chance of survival over a period of 5 years. However, the study showed that patients who were successfully weaned from PMV and discharged had the highest five-year survival rate. We found that successfully weaned from mechanical ventilation and receiving a tracheostomy were two important factors associated with improved long-term survival outcomes in patients requiring prolonged mechanical ventilation.

## Figures and Tables

**Figure 1 medicina-59-00614-f001:**
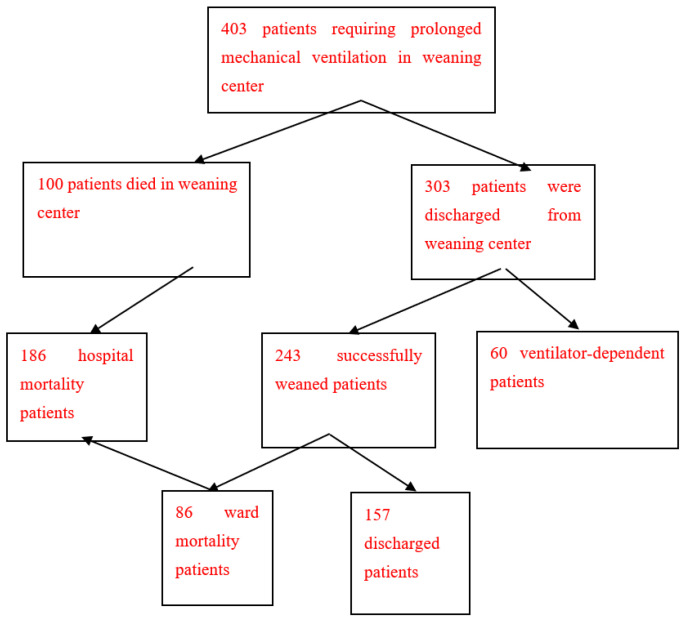
Clinical outcomes of 403 patients requiring prolonged mechanical ventilation. 1. The long-term follow-up data of 403 prolonged mechanical ventilation patients included 100 respiratory care center mortality prolonged mechanical ventilation patients and 303 patients discharged from respiratory care center. 2. Among the 303 patients discharged from the respiratory care center, 243 were successfully weaned from prolonged mechanical ventilation and 60 were ventilator-dependent. 3. One hundred and eighty-six patients who died in the hospital after prolonged mechanical ventilation included 100 patients who died in the respiratory care center and 86 successfully weaned patients who died in the ward. 4. Two hundred and forty-three prolonged mechanical ventilation patients were successfully weaned, including 157 discharged prolonged mechanical ventilation patients and 86 ward mortality prolonged mechanical ventilation patients.

**Figure 2 medicina-59-00614-f002:**
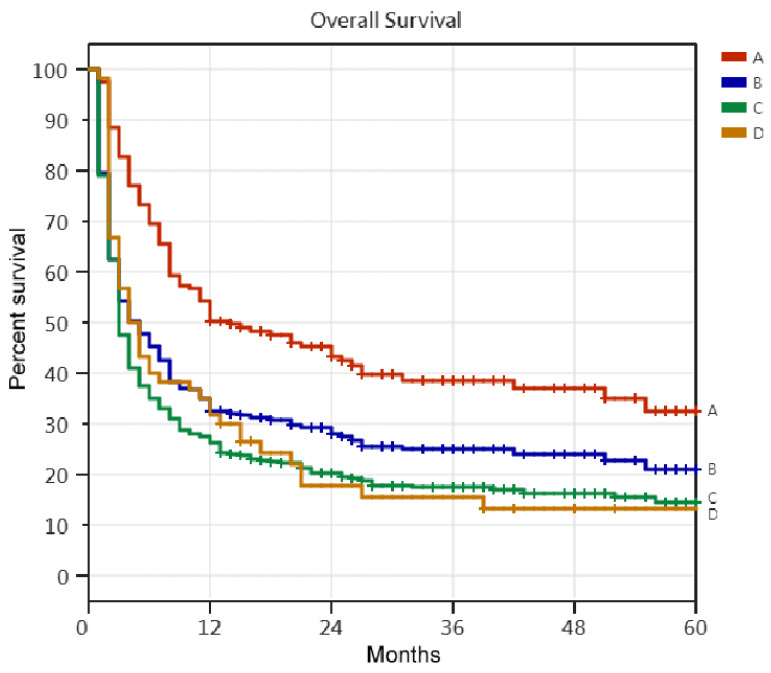
The Kaplan–Meier curve of prolonged mechanical ventilation patients, successfully weaned prolonged mechanical ventilation patients, patients discharged after successfully weaned from prolonged mechanical ventilation, and respiratory care word patients. 1. A: 157 discharged PMV patients: B: 243 successfully weaned PMV patients; C: 403 PMV patients; D: 60 respiratory care ward patients. 2. 157 discharged PMV patients vs. 403 PMV patients: *p* < 0.001, HR = 2.166; 157 discharged PMV patients vs. 60 RCW patients: *p* < 0.001, HR = 1.965; 157 discharged PMV patients vs. 243 successfully weaned PMV patients: *p* < 0.001, HR = 1.658; 60 RCW patients vs. 403 PMV patients: *p* = 0.148, HR = 1.246; 60 RCW patients vs. 243 successfully wended PMV patients: *p* = 0.575, HR = 0.920; 243 successfully weaned PMV patients vs. 403 PMV patients: *p* = 0.001, HR = 1.374. 3. The study found that the five-year survival rate of respiratory care ward patients was similar to that of patients who had been successfully weaned from prolonged mechanical ventilation. The study also found that the 157 patients who were discharged after being successfully weaned from prolonged mechanical ventilation had the best five-year survival rate among the study participants.

**Table 1 medicina-59-00614-t001:** Clinical variables and tracheostomy of 403 patients requiring prolonged mechanical ventilation.

	Ward Mortality Patients	Discharged PMV Patients	RCC Mortality Patients	RCW Patients
	(No = 86)	(No = 157)	(No = 100)	(No = 60)
Tracheostomy, No	4 (4.65%)	29 (18.47%)	12 (12.0%)	20 (33.3%)
Age Groups, No				
Age < 45 Y/O, No	3 (3.49%)	4 (2.55%)	1 (1.0%)	1 (1.67%)
Age 45–54 Y/O, No	6 (6.98%)	14 (8.92%)	9 (9.0%)	9 (15.0%)
Age 55–64 Y/O, No	7 (8.14%)	22 (14.01%)	14 (14.0%)	4 (6.67%)
Age 65–74 Y/O, No	22 (25.58%)	36 (22.93%)	15 (15.0%)	15 (25.0%)
Age 75–84 Y/O, No	34 (39.53)	51 (32.48%)	42 (42.0%)	26 (43.3%)
Age ≥ 85 Y/O, No	14 (16.28%)	30 (19.11%)	19 (19.0%)	5 (8.33%)
Causes of respiratory failure led to PMV, No				
pneumonia, No	31 (36.04%)	65 (41.4%)	34 (34.0%)	19 (31.7%)
intracranial hemorrhage, No	20 (23.26%)	31 (19.75%)	13 (13.0%)	4 (6.67.%)
sepsis, No	10 (11.63%)	11 (7.01%)	15 (15.0%)	9 (15.0%)
COPD, No	2 (2.33%)	7 (4.46%)	7 (7.0%)	7 (11.6%)
cardiac disease, No	6 (6.98%)	8 (5.1%)	8 (8.0%)	3 (5.0%)
malignant patients, No	2 (2.33%)	3 (1.91%)	7 (7.0%)	3 (5.0%)
post operation, No	3 (3.49%)	12 (7.64%)	6 (6.0%)	7 (11.6%)
cervical spine diseases, No	0(0%)	3 (1.91%)	2 (2.0%)	1 (1.67%)
post-CPCR, No	2 (2.33%)	2 (1.27%)	1 (1.0%)	2 (3.33%)
cerebral infarction, No	3 (3.49%)	3 (1.91%)	1 (1.0%)	0 (0%)
Comorbidity, No				
cardiovascular disease, No	58 (67.44%)	100 (63.7%)	65 (65.0%)	37 (61.7%)
chronic lung disease, No	13 (15.12%)	36 (22.93%)	19 (19.0%)	12 (2.0%)
chronic kidney disease, No	16 (18.6%)	16 (10.19%)	12 (12.0%)	5 (8.33%)
end-stage renal disease (requiring dialysis), No	12 (13.95%)	9 (5.73%)	11 (11.0%)	5 (8.33%)
neurologic disease, No	27 (31.4%)	52 (33.12%)	27 (27.0%)	21 (35.0%)
metabolic disease, No	39 (45.35%)	56 (35.67%)	35 (35.0%)	27 (45.0%)
malignant diseases, No	15 (17.44%)	16 (10.19%)	26 (26.0%)	6 (10.0%)
no comorbidity, No	4 (4.65%)	19 (12.1%)	4 (4.0%)	4 (6.67%)
one comorbidity, No	25 (29.07%)	33 (21.02%)	30 (30.0%)	19 (31.7%)
two comorbidities, No	27 (31.4%)	53 (33.76%)	30 (30.0%)	15 (25.0%)
three comorbidities, No	14 (16.28%)	40 (25.48%)	25 (25.0%)	17 (28.3%)
≥four comorbidities, No	16 (18.6%)	12 (7.64%)	11 (11.0%)	5 (8.33%)

No: number, Y/O: years old, PMV: prolonged mechanical ventilation, COPD: chronic obstructive pulmonary disease, CPCR: cardiopulmonary–cerebral resuscitation, chronic lung disease (such as chronic obstructive pulmonary disease, asthma, bronchiectasis, pulmonary fibrosis, interstitial lung disease).

**Table 2 medicina-59-00614-t002:** The survival time of 403 prolonged mechanical ventilation patients.

Patients, No /Survival Time	Patients Requiring PMV (No = 403)	Successfully Weaned Patients (No = 243)	Discharged PMV Patients (No = 157)	RCW Patients (No = 60)
Dead patients, No	331 (82.1%)	181 (74.5%)	95 (60.5%)	50 (83.3%)
0–3 months, No	212 (52.6%)	111 (45.7%)	27 (17.2%)	26 (43.3%)
4–6 months, No	50 (12.4%)	22 (9.0%)	21 (13.4%)	10 (16.7%)5(8.3%)
7–12 months, No	35 (8.7%)	31 (12.8%)	30 (19.1%)	7 (11.7%)
less than 2 years, No	22 (5.5%)	9 (3.7%)	9 (5.7%)	1 (1.7%)
less than 3 years, No	8 (2.0%)	5 (2.0%)	5 (3.2%)	1 (1.7%)
less than 4 years, No	2 (0.5%)	1 (0.4%)	1 (0.6%)	0
less than 5 years, No	2 (0.5%)	2 (0.8%)	2 (1.3%)	0
more than 5 years, No	0	0	0	
Alive patients, No	72 (17.9%)	62 (25.5%)	62 (39.5%)	0
0–3 months, No	0	0	0	0
4–6 months, No	0	0	0	0
7–12 months, No	1 (0.30%)	1 (0.4%)	1 (0.6%)	4 (6.7%)
less than 2 years, No	25 (6.2%)	21 (8.6%)	21 (13.4%)	0
less than 3 years, No	10 (2.5%)	10 (4.1%)	10 (6.7%)	3 (5.0%)
less than 4 years, No	14 (3.5%)	11 (4.5%)	11 (7.0%)	1 (1.7%)
less than 5 years, No	13 (3.2%)	12 (4.9%)	12 (7.6%)	2 (3.3%)
more than 5 years,	9 (2.2%)	7 (2.9%)	7 (4.5%)	

No: number, PMV: prolonged mechanical ventilation, RCW: respiratory care ward.

**Table 3 medicina-59-00614-t003:** The one-year survival of patients requiring prolonged mechanical ventilation.

Authors	Patients No.	One-Year Survival Rate (%)	Factors of Poor One-Year Survival Rate	Factors of Good One-Year Survival Rate
Stoller [9]	162	43%	older age	
Pilcher [10]	153	58%	older age, high	
			APACHE score	
Scheinhorn [11]	1419	40%	failure to wean	
Bigatello [12]	146	61%		
Cox [13]	126	56%		
Carson [14]	260	52%	age > 65 Y/O,	
			thrombocytopenia,	
			use vasopressors,	
Rose [15]	115	50%	ESRD	
Lin [16]	533	37.20%		young age, absence of liver cirrhosis
				tracheostomy
Huang [17]	401	46%		patients
			high APACHE II	
			score, CHF, ESRD,	
			malignancy, liver	no comorbidity
Huang [2]	403	24.30%	cirrhosis	
			age > 75Y/O, ESRD,	
			four comorbidities,	

No: number, Y/O: years old, APACHE: Acute Physiology and Chronic Health Evaluation, ESRD: end-stage renal disease, CHF: congestive heart failure.

**Table 4 medicina-59-00614-t004:** The one-year survival of patients discharged after successfully weaned from prolonged mechanical ventilation.

Authors	Patients No.	One-Year Survival Rate (%)	Factors of Poor One-Year Survival Rate	Factors of Good One-Year Survival Rate
Schonhofer [18]	293	49.40%	failure to wean	young age, low
				APACHE II score
Su [19]	244	44.60%		
Davies [20]	458	65.00%	failure to wean,	nocturnal use of
			older age,	NIV
Jubran [21]	315	66.90%	high SAPS score, high APACHII, low body mass index, failure to wean	
Warnke [22]	597	66.50%	older age,	
			failure to wean	
Huang [23]	157	50.30%		
			age > 85Y/O,	no comorbidity, undergoing tracheostomy
			four comorbidities,	

No: number, Y/O: years old, APACHE: Acute Physiology and Chronic Health Evaluation, NIV: Noninvasive mechanical Ventilation, SAPS: Simplified Acute Physiology Score.

## Data Availability

The datasets that were used and/or analyzed for this study are available from the corresponding author upon reasonable request.

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
