# Peer review of "The Survival Outcomes of Patients Requiring Prolonged Mechanical Ventilation"

_medicina, 2023, doi:10.3390/medicina59030614_

Round 1

Reviewer 1 Report (Previous Reviewer 2)

No additional comments

Author Response

Thank you for reviewing my article.

Reviewer 2 Report (New Reviewer)

This study has many overlaps with the another study of the author.(Huang C. How prolonged mechanical ventilation is a neglected disease in chest medicine: a study of prolonged mechanical ventilation based on 6 years of experience in Taiwan. Ther Adv Respir Dis. 2019 Jan-Dec;13:1753466619878552. doi: 10.1177/1753466619878552. PMID: 31566093; PMCID: PMC6769206.)

In authors section,  Huang and?? Not stated

In Table 1, Some causes of respiratory failure leading to PLR began with capital letters

Chronic lung disease should be listed, such as COPD, bronchiectasis.

Figure 1 should be clearer and elegant.

It was mentioned several times as ''in our previous study'',  it was not suitable.

Title of Table 2  written with small and big letters irregularly.

The term of no comorbidity rather than zero comorbidity is preferrable.

Section  about Covid and responsible pathogens is a separate topic and shouldn’t be discussed in this article. The patients included before  COVID-19 pandemic in this study.

Author Response

1.This study has many overlaps with the another study of the author.(Huang C. How prolonged mechanical ventilation is a neglected disease in chest medicine: a study of prolonged mechanical ventilation based on 6 years of experience in Taiwan. Ther Adv Respir Dis. 2019 Jan-Dec;13:1753466619878552. doi: 10.1177/1753466619878552. PMID: 31566093; PMCID: PMC6769206.)

Reply: The article (How prolonged mechanical ventilation is a neglected disease in chest medicine: a study of prolonged mechanical ventilation based on 6 years of experience in Taiwan) is reference 2 of this study. The hospital information, patient information, comorbidities, causes of respiratory failure leading to PMV, and RCC admission eligibility criteria were identical to those of the reference 2. The reference 2 did not explored the issue of five-year survival rate of patients requiring prolonged mechanical ventilation.

2.In authors section, Huang and?? Not stated

Reply: I corrected.

3.In Table 1, Some causes of respiratory failure leading to PLR began with capital letters

Reply: I corrected.

4.Chronic lung disease should be listed, such as COPD, bronchiectasis.

Reply: chronic lung disease (such as chronic obstructive pulmonary disease

, asthma, bronchiectasis, pulmonary fibrosis, interstitial lung disease)

5.Figure 1 should be clearer and elegant.

Reply: I corrected.

6.It was mentioned several times as ''in our previous study'',  it was not suitable.

Reply: I corrected.

7.Title of Table 2 written with small and big letters irregularly.

Reply: I corrected.

8.The term of no comorbidity rather than zero comorbidity is preferrable.

Reply: I corrected.

9.Section about Covid and responsible pathogens is a separate topic and shouldn’t be discussed in this article. The patients included before COVID-19 pandemic in this study.

Reply:

1.Based on reviewer input and editorial evaluation, the editor encourage resubmission of my manuscript (manuscript ID [medicina-1881925]) after extensive revisions.

2.Previous reviewer 1 suggest that “Please improve these sentences by inserting a short paragraph on COVID 19”

3.Previous reviewer 2 suggest that “The long-term prognosis of mechanical ventilation patients is a topic of interesting, especially in the COVID-19 era.”

Therefore, I discuss the issue “prolonged mechanical ventilation of COVID-19–related acute hypoxemic respiratory failure and responsible pathogens”

Reviewer 3 Report (New Reviewer)

I would like to congratulate the authors on their work, however I have some comments concerning the study.

General Comments:

First and foremost, the timeline of the study is unclear. If I understand correctly the essence of the current study than the study attempted to collect patients with prolonged mechanical ventilation (greater than 21 days) and follow them from that point for five years. Patients were recruited up to the end of 2018 – therefore some of them were not followed up for five years. This has to be clarified throughout.

Secondly as the inclusion period is up to 2018, by definition, no patients with COVID19 were included. The only possibility is that some of those long term post mechanical ventilation patients contracted COVID as a secondary infection. I think that as all of the published data on COVID (certainly all that is referred to in the current study) relates to COVID as the primary disease adding the data on COVID in survivors of previous PMV has to be done very cautiously and only in that specific context.

It is not clear whether this study relates to the same patients as in ref (2) or to a different group of patients

The print layout of the article is not consistent and confusing

Specific comments:

Extensive English editing is required.

I would avoid using all of the different abbreviations used in the paper, I found them confusing.

The author's name as appears in the manuscript is "Chienhsiu Huang and" I presume this is incorrect

First paragraph of introduction should start with the placing of the prolonged respiratory care service structure (In Taiwan, in our hospital…)

I quote “we have published 6 papers… “ but they are not referred or discussed. The number of previous papers is not important

I do not see why discuss COVID except for maybe briefly saying that the patients are all preCOVID

Methods:

Data collection – again is it similar to ref No 2? Dates here include until the end of 2017. I t states here that data was reviewed for long term survival until the end of 2018 (do I understand correctly?) If so then again the timeline is very ambiguous and requires clarification (because most patients had a follow-up period of much less than 5 years) If all of the follow up data is until the end of 2018 than you can omit everything about COVID as it is completely out of the scope of this paper.

Results:

Defining the different groups of patients according to the various facilities remains unclear. I would add a flow-chart showing the course of the patient. Similar to that of figure 1 but with more emphasis on the time course.

Figure 1 is graphically un clear (boxes over one another

Table 1 why no comparison between groups?

Need to elaborate on the low incidence of tracheostomies. How are patients who require chronic ventilation treated?

Discussion:

Page 6 of 16 line 172: What is meant by “high chronic health evaluation II” this is not APACHE II

Line 175: In contrast is incorrect. You also found older patients died more frequently

Line 182 – what is meant by “excellent”

Page 8 of 16 (Five year survival)

Line 214-216: as expected, patients who are weaned of MV have a better long term outcome than those who are not.

However in lines 221-225 you state that long term outcome was similar between those weaned and those on chronic ventilation (how many of the group of 60 were actually followed up for 5 years?)

Page 9 of 16:

Paragraph beginning in line 240 (ref 23) It is not clear if the same patients (or some of them) are included in both studies

All data on COVID is irrelevant

Author Response

Dear reviewer:

A more detailed reply is in a word file.

1.First and foremost, the timeline of the study is unclear. If I understand correctly the essence of the current study than the study attempted to collect patients with prolonged mechanical ventilation (greater than 21 days) and follow them from that point for five years. Patients were recruited up to the end of 2018 – therefore some of them were not followed up for five years. This has to be clarified throughout.

Reply:

  1. We retrospectively studied all of the medical records of 574 patients who were admitted to the RCC from January 2012 to December 2017.
  2. We reviewed all PMV patients' outpatient medical records up until December 31, 2018.
  3. The five-year survival rate was calculated using the Kaplan-Meier estimate of the survivor method. As a result, some of them were not follow-up with for five years. 

2.Secondly as the inclusion period is up to 2018, by definition, no patients with COVID19 were included. The only possibility is that some of those long term post mechanical ventilation patients contracted COVID as a secondary infection. I think that as all of the published data on COVID (certainly all that is referred to in the current study) relates to COVID as the primary disease adding the data on COVID in survivors of previous PMV has to be done very cautiously and only in that specific context.

Reply:

1.Based on reviewer input and editorial evaluation, the editor encourage resubmission of my manuscript (manuscript ID [medicina-1881925]) after extensive revisions.

2.Previous reviewer 1 suggest that “Please improve these sentences by inserting a short paragraph on COVID 19”

3.Previous reviewer 2 suggest that “The long-term prognosis of mechanical ventilation patients is a topic of interesting, especially in the COVID-19 era.”

Therefore, I discuss the issue “prolonged mechanical ventilation of COVID-19–related acute hypoxemic respiratory failure and responsible pathogens

3.It is not clear whether this study relates to the same patients as in ref (2) or to a different group of patients

Reply: This study relates to the same patient populations as the reference (2), and it is a continuation of research on a different topic related to the five-year survival rate of patients requiring prolonged mechanical ventilation. The reference 2 did not explored the issue of five-year survival rate of patients requiring prolonged mechanical ventilation.

4.The print layout of the article is not consistent and confusing

Reply: I corrected, to the best of my ability.

Specific comments:

5.Extensive English editing is required.

Reply: The manuscript has been carefully reviewed by Charlesworth Author Services.

6.I would avoid using all of the different abbreviations used in the paper, I found them confusing.

Reply: I corrected.

7.The author's name as appears in the manuscript is "Chienhsiu Huang and" I presume this is incorrect

Reply: This is correct.

8.First paragraph of introduction should start with the placing of the prolonged respiratory care service structure (In Taiwan, in our hospital…)

Reply: I corrected.

In 2004, the National Association for Medical Direction of Respiratory Care held a conference associated with the care and management of patients requiring prolonged mechanical ventilation (PMV). Patients on PMV are defined as those who use a mechanical ventilator for at least 6 hours daily for at least 21 consecutive days [1]. We reported the clinical experience of 574 patients requiring PMV, and our patients presented an abysmal 1-year rate (24.3%) in 2019 [2]. A study by Carson showed that patients requiring PMV had poor long-term outcomes, which had not appreciably improved in 2006 [3]. By 2012, Carson reported that the 1-year survival rate of PMV patients was 52%, which is an insufficient increase in survival time [4]. According to the research by Damuth et al., the overall 1-year survival rate for weaned units in acute care hospitals was 45.2% [5].

The comprehensive care program for ventilator-dependent patients encompasses mechanical ventilator care in four settings: the intensive care unit (also known as the acute critical care stage), the respiratory care center (also known as the RCC) (the weaning center in the acute care hospital for weaning training), the respiratory care ward (also known as the RCW) (the ventilator-dependent patients long-term care ward), and home care services (a stable period in which the patient is cared for directly by family caregivers or nurses who work in nursing homes) in Taiwan [6]. The Dalin Tzu Chi Hospital is a tertiary-level teaching hospital with 600 acute-care beds and an intensive care unit containing 59 beds. A 10-bed ventilator weaning center (RCC) is available within Dalin Tzu Chi Hospital.

9.I quote “we have published 6 papers… “ but they are not referred or discussed. The number of previous papers is not important

Reply: I corrected.

We reported in 2019 the comprehensive clinical experience of those on PMV. The hospital information, patient information, comorbidities, causes of respiratory failure led to PMV, and RCC admission eligibility criteria were identical to those of studies [2].

  1. I do not see why discuss COVID except for maybe briefly saying that the patients are all preCOVID

Reply:

1.Based on reviewer input and editorial evaluation, the editor encourage resubmission of my manuscript (manuscript ID [medicina-1881925]) after extensive revisions.

2.Previous reviewer 1 suggest that “Please improve these sentences by inserting a short paragraph on COVID 19”

3.Previous reviewer 2 suggest that “The long-term prognosis of mechanical ventilation patients is a topic of interesting, especially in the COVID-19 era.”

Therefore, I discuss the issue “prolonged mechanical ventilation of COVID-19–related acute hypoxemic respiratory failure and responsible pathogens

11.Methods:

Data collection – again is it similar to ref No 6? Dates here include until the end of 2017. I t states here that data was reviewed for long term survival until the end of 2018 (do I understand correctly?) If so then again the timeline is very ambiguous and requires clarification (because most patients had a follow-up period of much less than 5 years) If all of the follow up data is until the end of 2018 than you can omit everything about COVID as it is completely out of the scope of this paper.

Reply:

  1. This study relates to the same patient populations as the reference (2), and it is a continuation of research on a different topic related to the five-year survival rate of patients requiring prolonged mechanical ventilation.
  2. The five-year survival rate calculated using the Kaplan-Meier estimate of the survivor method. As a result, some of them were not follow-up with for five years.
  3. Based on reviewer input and editorial evaluation, the editor encourage resubmission of my manuscript (manuscript ID [medicina-1881925]) after extensive revisions. Therefore, I discuss the issue “prolonged mechanical ventilation of COVID-19–related acute hypoxemic respiratory failure and responsible pathogens

12.Results:

Defining the different groups of patients according to the various facilities remains unclear. I would add a flow-chart showing the course of the patient. Similar to that of figure 1 but with more emphasis on the time course. Figure 1 is graphically unclear (boxes over one another)

Reply: I give a very detailed explanation in Figure 1 legend.

1.The long-term follow-up data of 403 prolonged mechanical ventilation patients included 100 respiratory care center mortality prolonged mechanical ventilation patients and 303 patients discharged from respiratory care center.

2.Among the 373 patients discharged from the respiratory care center, 243 were successfully weaned from prolonged mechanical ventilation and 60 were ventilator-dependent.

3.One hundred and eighty-six patients who died in the hospital after prolonged mechanical ventilation included 100 patients who died in the respiratory care center and 86 patients who died in the ward.

4.Two hundred and forty-three prolonged mechanical ventilation patients were successfully weaned, including 157 discharged prolonged mechanical ventilation patients and 86 ward mortality prolonged mechanical ventilation patients.

13.Table 1 why no comparison between groups?

Reply: Table 1 showed basic clinical variables and tracheostomy of 403 patients requiring prolonged mechanical ventilation. We focus on five-year survival rate and the comparison between groups show in Figure 2.

14.Need to elaborate on the low incidence of tracheostomies. How are patients who require chronic ventilation treated?

Reply: For RCC patients who cannot be weaned from the ventilator in the short term, a tracheostomy is recommended. However, the majority of patients and family members are opposed to the procedure. The most common reasons for refusal are concern that the operation will leave a wound on the patient's neck, concern about the risks and complications of tracheostomy. Dalin Tzu Chi Hospital implemented the program "If my family has difficulty weaning from the ventilator, does he/she need tracheostomy?" in 2017. During the three-year study period, only 37 PMV patients (9.7%) underwent tracheostomy. The clinical situation of PMV patients receiving or not receiving tracheostomy in Taiwan differs from that of Western countries.

Discussion:

  1. Page 6 of 16 line 172: What is meant by “high chronic health evaluation II” this is not APACHE II

Reply: I change to Acute Physiology and high chronic health evaluation II.

  1. Line 175: In contrast is incorrect. You also found older patients died more frequently

Reply: I correct to “We also found that patients aged >75 years had a poorer 1-year survival rate.”

17.Line 182 – what is meant by “excellent”

Reply: I correct to “while PMV patients who had undergone tracheostomy have a favorable one-year survival rate “

18.Page 8 of 16 (Five-year survival) Line 214-216: as expected, patients who are weaned off MV have a better long-term outcome than those who are not.

Reply: Yes, as expected.

19.However in lines 221-225 you state that long term outcome was similar between those weaned and those on chronic ventilation (how many of the group of 60 were actually followed up for 5 years?)

Reply: I follow up on these RCW patients until December 31, 2018, and there are eight patients whose follow-up time is less than five years. Fifty patients died in less than five years, and two patients’ follow-up time was more than five years.

20.Page 9 of 16: Paragraph beginning in line 240 (ref 23) It is not clear if the same patients (or some of them) are included in both studies

Reply: The study of reference 23 ( same patients requiring PMV) showed a multivariate analysis comparing the clinical variables and receipt or not of tracheostomy between 86 ward mortality PMV patients and 157 discharged PMV patients (same patients of this study) showed that ESRD, receipt or not of tracheostomy, and ≥4 comorbidities were significantly different (table 3).

21.All data on COVID is irrelevant

Reply:

1.Based on reviewer input and editorial evaluation, the editor encourage resubmission of my manuscript (manuscript ID [medicina-1881925]) after extensive revisions.

2.Previous reviewer 1 suggest that “Please improve these sentences by inserting a short paragraph on COVID 19”

3.Previous reviewer 2 suggest that “The long-term prognosis of mechanical ventilation patients is a topic of interesting, especially in the COVID-19 era.”

Therefore, I discuss the issue “prolonged mechanical ventilation of COVID-19–related acute hypoxemic respiratory failure and responsible pathogens

Round 2

Reviewer 2 Report (New Reviewer)

-

Author Response

Reply:

1.The manuscript has been carefully reviewed by Charlesworth Author Services before.

2.The manuscript has been carefully reviewed by my English teach on March 10, 2023.

Reviewer 3 Report (New Reviewer)

General Comments:

I asked the Editor to decide upon the importance of the data on COVID.

If you write as a single author than maybe I should be used rather than “we”

The methods section has to be structured

Odds ratio should be accompanied by confidence intervals

Specific comments:

Introduction:

 First and second sentences can be combined.

Line 84 – instead of “abysmal” I would use a more neutral word such as “poor” or “low”

Line 87: delete “time” after survival

Line 90: “The comprehensive care program for ventilator-dependent patients in Taiwan encompasses” move in Taiwan to the beginning rather than the end of the sentence.

After line 115:  add – In the present study we set out to evaluate the long-term survival of these patients based on data collected until the end of 2018 (if I understand correctly)

Again as there is no data concerning COVID there is no point to mentioning it in the introduction.

A sentence can be added to the discussion saying that s the data were collected prior to COVID the author suggests that long term follow up of ventilated COVID patients is an interesting topic which is beyond the scope of this article. (again at the Editors discretion, I understand you worked hard for this)

Methods:

There is no clear description of the population studied (it is defined in the introduction)

There is no clear description of the main and secondary outcomes (again, presented in the introduction)

Results: How long was the mean follow-up period.

How many patients have been included for each year of long term follow-up (same as for figure 2).

Page 9/32 line 155: The corresponding rates for 243 successful weaning from PMV patients were 32.5%,  28.0%, 29.4%, 24.0%, and 21.0%, respectively.

How can you have an increase in survival over time?

Discussion:

4.1

At present: “The one-year survival rate of the 403 PMV patients in our series was 24.3%. In our study indicated that PMV patients with no comorbidities (P = 0.002, odds ratio [OR]”

I suggest instead of “in our study indicated that…” “We found that...”

Figure 2: Have to include the number of patients still alive after each year of the study as by definition most of the patients did not have 5 year survival assessed.

Author Response

1.I asked the Editor to decide upon the importance of the data on COVID.

Reply: Let Editor make a decision.

2.If you write as a single author than maybe I should be used rather than “we”

Reply: I correct.

3.The methods section has to be structured

Reply: 2. Methods: 2.1 Setting and Participants; 2.2 Data collection; 2.3 Outcomes Measure; 2.4 Statistical analysis.

4.Odds ratio should be accompanied by confidence intervals

Reply: I correct.

Specific comments:

Introduction:

5.First and second sentences can be combined.

Reply: I correct.

6.Line 84 – instead of “abysmal” I would use a more neutral word such as “poor” or “low”

Reply: I correct to “poor”..

7.Line 87: delete “time” after survival

Reply: I correct.

8.Line 90: “The comprehensive care program for ventilator-dependent patients in Taiwan encompasses” move in Taiwan to the beginning rather than the end of the sentence.

Reply: I correct.

9.After line 115:  add – In the present study we set out to evaluate the long-term survival of these patients based on data collected until the end of 2018 (if I understand correctly)

Reply: I add.

10.Again as there is no data concerning COVID there is no point to mentioning it in the introduction. A sentence can be added to the discussion saying that the data were collected prior to COVID the author suggests that long term follow up of ventilated COVID patients is an interesting topic which is beyond the scope of this article. (again at the Editors discretion, I understand you worked hard for this)

Reply: I add.

Methods:

11.There is no clear description of the population studied (it is defined in the introduction)

Reply:

2.1 Setting and Participants:

The Dalin Tzu Chi Hospital is a tertiary-level teaching hospital with 600 acute care beds and an ICU containing 59 beds. A 10-bed ventilator weaning unit is available within Dalin Tzu Chi Hospital which is a weaning unit within an acute-care hospital, provides care for patients on PMV. I conducted a retrospective, single-center study, enrolled all patients consecutively admitted to the RCC of Dalin Tzu Chi Hospital between January 1, 2012, and December 31, 2017. Patients were eligible for RCC admission if they met the Taiwan national health insurance requirements: (a) hemodynamic stability; (b) no vasoactive drug infusion needed; (c) stable oxygen condition (O2 saturation ⩾ 90% or PaO2 ⩾ 60 mmH) with the requirement fraction of inspired oxygen less than 40% and positive end-expiratory pressure less than10 cm H2O; (d) no critical acute hepatic or renal failure; (e) no massive upper gastrointestinal bleeding; (f) no requirement for surgical intervention within the ensuing 2 weeks or no large open surgical wound; (g) no uncontrolled severe infectious diseases; (h) no life-threatening arrhythmia.

12.There is no clear description of the main and secondary outcomes (again, presented in the introduction)

Reply: The main outcome is to investigate the five-year survival rate of prolonged mechanical ventilation patients using the Kaplan-Meier estimate of the survivor method. The secondary outcome is to investigate the factors that influence five-year survival rate among all PMV patients, successfully weaned PMV patients, patients discharged after successfully weaned PMV patients (discharged PMV patients), and ventilator-dependent PMV patients (RCW patients).

  1. Results: How long was the mean follow-up period. How many patients have been included for each year of long term follow-up (same as for figure 2).

Reply: A new table 2.

  1. Page 9/32 line 155: The corresponding rates for 243 successfully weaned from PMV patients were 32.5%, 28.0%, 29.4%, 24.0%, and 21.0%, respectively. How can you have an increase in survival over time?

Reply: The 243 successfully weaned from PMV included 157 discharged PMV patients and 86 ward mortality patients. The factors between ward mortality patients and those discharged PMV patients revealed that the poorer survival of ward mortality patients were due to a higher percentage of end-stage renal disease comorbidity, a higher percentage of ≥4 comorbidities and a lower percentage of undergoing tracheostomy.

The receipt or not of tracheostomy is the key influential factor of long-term survival of successfully weaned prolonged mechanical ventilation patients. Tracheostomy should be attempted in suitable patients for improving long-term outcomes in successfully weaned prolonged mechanical ventilation patients. (reference 23)

Discussion:

4.1

At present: “The one-year survival rate of the 403 PMV patients in our series was 24.3%. In our study indicated that PMV patients with no comorbidities (P = 0.002, odds ratio [OR]” I suggest instead of “in our study indicated that…” “We found that...”

Reply: I correct.

Figure 2: Have to include the number of patients still alive after each year of the study as by definition most of the patients did not have 5 year survival assessed.

Reply: a new Table 2.

This manuscript is a resubmission of an earlier submission. The following is a list of the peer review reports and author responses from that submission.

Round 1

Reviewer 1 Report

I read this article with pleasure, relating to a topic that is certainly of interest. Here are some suggestions for possible improvements:

Abstract: The content is complete and clear, easy to understand on first reading.

Introduction: I would specify if this study considers patients with the same characteristics as studies prior to this and mentioned in this article did "We reported clinical experience of 574 patients receiving PMV in 2019, and our PMV patients presented an abysmal 1-year survival rate (24.3%) [2]. A study by Carson showed that patients requiring PMV have poor long-term outcomes, which have not improved significantly in 2006 [3]. By 2012, Carson reported the 1-year survival rate of prolonged mechanical ventilation patients was 52%, an unsatisfactory improvement in survival time [4]. Damuth et al. reported a 1-year survival rate of 45.2% across all weaned units in acute care hospitals [5]. No study has explored the 5-year survival rate of PMV patients. Please improve these sentences by inserting a short paragraph on COVID 19 and add these articles: a- Different Methods to Improve the Monitoring of Noninvasive Respiratory Support of Patients with Severe Pneumonia/ARDS Due to COVID-19: An Update. J. Clin. Med. 2022, 11, 1704. https://doi.org/10.3390/jcm11061704 b- Bedside Selection of Positive End Expiratory Pressure by Electrical Impedance Tomography in Patients Undergoing Veno-Venous Extracorporeal Membrane Oxygenation Support: A Comparison between COVID-19 ARDS and ARDS from Other Etiologies. J. Clin. Med. 2022, 11, 1639. https://doi.org/10.3390/jcm11061639 Methods: I would replace "Definitions" chapter in "Introduction" section, providing if possible further details on management of PMV in terms of used tools and any concomitant therapies. Regarding chapter "Data collection", I would provide some details on the characteristics of the patients included in the study. Were there any exclusion or inclusion criteria? Discussion: For a better reading of results, I would insert a table that distinguishes conditions that allowed a better or worse one-year survival rate in all PMV patients. Limitations of our study: I ​​appreciated the presence of this section. Conclusion: this section is clear and complete.

Reviewer 2 Report

The long-term prognosis of mechanical ventilation patients is a topic of interesting, especially in the COVID-19 era. However, the study has some major flaws, thus i have to recommend rejection for it. Firstly, the cause of  mechanical ventilation should be presented. It is well known that the internal and surgical ventilated patients have different clinical outcomes.  Secondly, the basic characteristic of included patients should also be presented. For example, older age is a known risk factor for poor prognosis, so this parameter should be adjusted to make the conclusion more reliable. Thirdly, this article has many grammar mistakes and is hard to read. For example, in the abstract, "our aim to improve survival outcomes of prolonged mechanical ventilation patients" , i can't find the predicate verb.